# Compensatory sequence variation between *trans*-species small RNAs and their target sites

**Nathan R Johnson[1,2], Claude W dePamphilis[1,2], Michael J Axtell[1,2]***

[1]Intercollege PhD Program in Plant Biology, Huck Institutes of the Life Sciences, The Pennsylvania State University, University Park, United States; [2]Department of Biology, The Pennsylvania State University, University Park, United States

**Abstract** *Trans*-species small regulatory RNAs (sRNAs) are delivered to host plants from diverse pathogens and parasites and can target host mRNAs. How *trans*-species sRNAs can be effective on diverse hosts has been unclear. Multiple species of the parasitic plant *Cuscuta* produce *trans*-species sRNAs that collectively target many host mRNAs. Confirmed target sites are nearly always in highly conserved, protein-coding regions of host mRNAs. *Cuscuta trans*-species sRNAs can be grouped into superfamilies that have variation in a three-nucleotide period. These variants compensate for synonymous-site variation in host mRNAs. By targeting host mRNAs at highly conserved protein-coding sites, and simultaneously expressing multiple variants to cover synonymous-site variation, *Cuscuta trans*-species sRNAs may be able to successfully target multiple homologous mRNAs from diverse hosts.

*For correspondence:
mja18@psu.edu

**Competing interests:** The authors declare that no competing interests exist.

## Introduction

Small regulatory RNAs (sRNAs) produced in one organism can sometimes function to silence mRNAs in another organism. These *trans*-species sRNAs seem especially prominent in plant/pathogen and plant/parasite interactions. Fungal plant pathogens produce sRNAs with complementarity to host mRNAs (*Weiberg et al., 2013*) and host plants produce *trans*-species sRNAs that silence mRNAs in both pathogenic fungi (*Zhang et al., 2016*; *Cai et al., 2018*) and oomycetes (*Hou et al., 2019*). The parasitic plant *Cuscuta campestris* produces *trans*-species microRNAs (miRNAs) which silence mRNAs in multiple host plants (*Shahid et al., 2018*). Silencing by plant *trans*-species sRNAs relies on extensive complementarity between the sRNA and target mRNA, similar to normal endogenous plant miRNAs (*Liu et al., 2014*).

Trans-species silencing is expected to benefit the source organism while being detrimental to the target organism in parasitic/pathogenic relationships. This implies that target sites are not under purifying selection to maintain complementarity to *trans*-species sRNAs. How could such a system be stable over evolutionary time and/or be useful against multiple species? One suggestion is a 'shotgun' strategy, in which a very diverse set of *trans*-species sRNAs is deployed to hit target mRNAs randomly. The plant response to *Phytophthora* may make use of this strategy (*Hou et al., 2019*). However, the fact that the *trans*-species sRNAs delivered to hosts from the parasitic plant *C. campestris* are miRNAs (*Shahid et al., 2018*) argues against the shotgun hypothesis in this case. MiRNAs are defined by the precise excision of a single mature, functional small RNA (*Axtell and Meyers, 2018*), which implies selection for the miRNA to target a particular sequence or closely related set of sequences. We examined *Cuscuta trans*-species sRNAs and their targets in detail to shed light on how this system may be evolutionarily stable and robust against diverse hosts.

# Results

We analyzed sRNA expression from four *Cuscuta* species (*Figure 1A*). Specimens from two or three distinct populations of *C. pentagona* and *C. gronovii*, respectively, were included, making a total of seven separate sRNA expression studies (identified with acronyms for brevity; *Figure 1A–B*, *Supplementary files 1–2*). All four *Cuscuta* species are generalists with documented hosts spanning multiple plant families (*Figure 1—figure supplement 1*). RNA samples (three biological replicates each) from host-parasite interfaces and parasite stems growing on the host *Arabidopsis thaliana* were obtained and used for sRNA sequencing (*Figure 1B*). Libraries were condensed to highly expressed sRNA variants and filtered to remove any sRNAs that came from the host (*Figure 1—figure supplement 2*). Differential expression analysis revealed several hundred *Cuscuta* sRNAs in each experiment that were significantly up-regulated in the interface tissue relative to parasite stems (FDR < 0.1) (*Supplementary file 3*); we dubbed these haustorially-induced (HI) sRNAs (*Figure 1A*; *Supplementary file 4*). HI-sRNAs are mostly 21 or 22 nucleotides long (*Figure 1A*), sizes consistent with either miRNAs or short interfering RNAs (siRNAs). Distinguishing miRNAs from siRNAs requires a genome assembly (*Axtell and Meyers, 2018*), a criterion met so far for only one of the four species (*C. campestris*) included in this study (*Shahid et al., 2018*; *Vogel et al., 2018*). Approximately half of the *C. campestris* HI-sRNAs (208/408) come from *MIRNA* hairpins (*Supplementary file 5*). *C. campestris*-derived HI-sRNAs were recovered from 40 of the 42 novel *MIRNA* loci described by

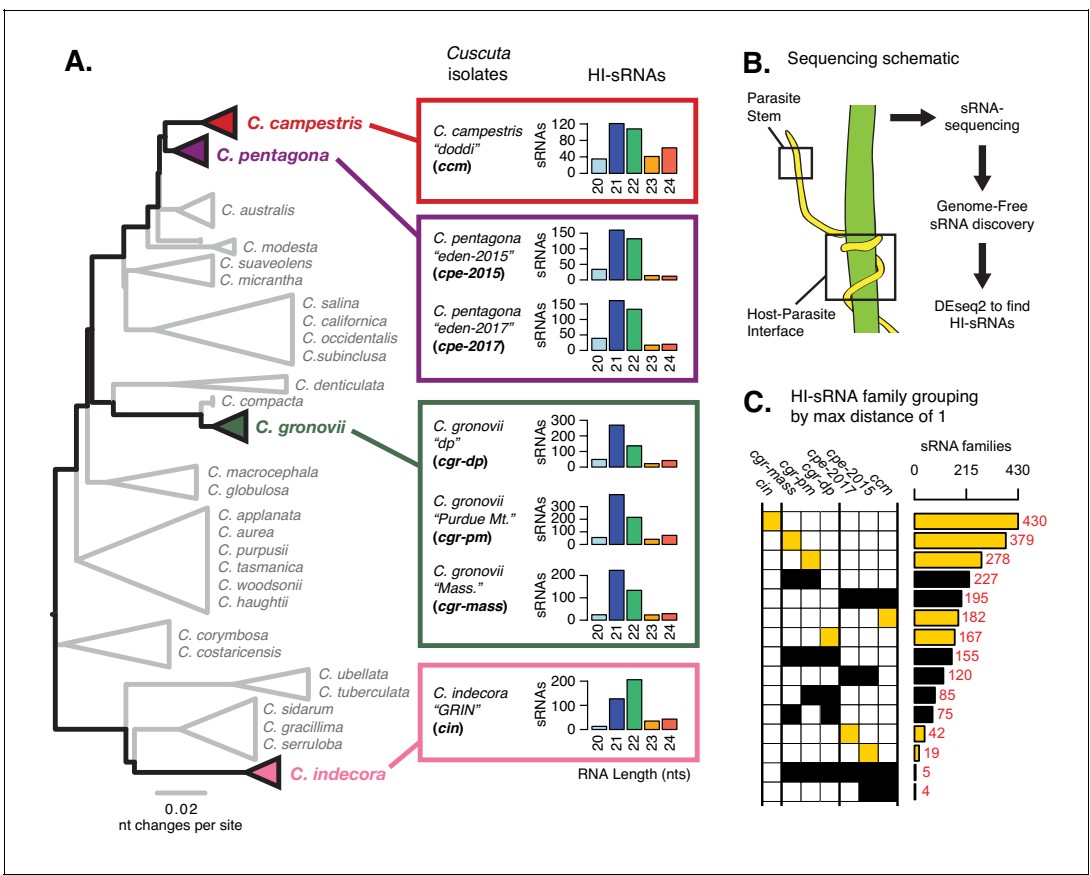

**Figure 1.** Haustorium-induced small RNAs (HI-sRNAs) are present in multiple *Cuscuta* species. (A) Phylogeny of select *Cuscuta* species. Size distribution of HI-sRNAs for each sequenced isolate and acronyms are shown. (B) Sampling and sequencing schematic to discern HI-sRNAs. (C) HI-sRNA family counts and membership for each isolate, showing only the top 15 groups. Families were grouped strictly using a maximum edit distance of one nucleotide. Yellow indicates families present in a single isolate.

The online version of this article includes the following figure supplement(s) for figure 1:

**Figure supplement 1.** Host preference in *Cuscuta* species in the United States.

**Figure supplement 2.** Genome-free HI-sRNA discovery pipeline.

*Shahid et al. (2018)*, including representatives from all five miRNA families previously demonstrated to target host mRNAs.

We next examined conservation of HI-sRNA accumulation between isolates and species. Some canonical plant miRNAs are highly conserved, with several ubiquitous families found in multiple plant orders, or even broadly in all land plants (*Cuperus et al., 2011*; *Chávez Montes et al., 2014*). Surprisingly, when using a strict cutoff (maximum edit distance of 1) we found that the majority of HI-sRNAs were not observed in more than one species (*Figure 1C*). In many cases HI-sRNAs were unique to single isolates of a single species (*Figure 1C*). This result implies that HI-sRNAs could be rapidly differentiating in expression, sequence, or both within these *Cuscuta* species.

Our previous work showed that *C. campestris* HI-sRNAs can target host mRNAs in several hosts (*Shahid et al., 2018*). We thus looked for evidence of interactions between our broader sets of HI-sRNAs with host (*A. thaliana*) mRNAs using two complementary methods: secondary siRNA accumulation (*Shahid et al., 2018*) and degradome analysis (*Addo-Quaye et al., 2008*). Secondary siRNAs can accumulate from mRNAs as a result of an initial miRNA- or siRNA-directed targeting event, especially when the initiating sRNA is 22 nucleotides long (*Cuperus et al., 2010*). A large portion of HI-sRNAs are 22-nt or are clustered from some sRNAs which are 22-nt in length (*Figure 1—figure supplement 2*), allowing us to detect their targeting with this approach. Degradome analysis made use of the NanoPARE method (*Schon et al., 2018*), which recovers 5′ ends of both capped and uncapped mRNAs. NanoPARE libraries were made from just one isolate from each of the four *Cuscuta* species, and comprised three biological replicates from the host portion of the interface (*Supplementary file 2*). *A. thaliana xrn4* mutants were used as hosts for these experiments because they over-accumulate 5′ remnants of sRNA-mediated mRNA cleavage (*Rymarquis et al., 2011*; *Schon et al., 2018*). The *CRCK2* mRNA is an example with both degradome and secondary siRNA evidence of targeting by a *C. pentagona* HI-sRNA (*Figure 2A–E*). Altogether these two analyses yielded a set of 61 target sites over 54 *A. thaliana* mRNAs targeted by *Cuscuta* HI-sRNAs confirmed by a single method and seven more confirmed by both (*Figure 2F*, *Figure 2—figure supplement 1A*, *Supplementary file 6*). Based on RNA-seq analysis, accumulation of confirmed target mRNAs is generally down-regulated in parasitized host stems (*Figure 3*). This greatly expands on the set of six mRNAs previously identified as host targets of *C. campestris* miRNAs (*Shahid et al., 2018*), and demonstrates that *trans*-species sRNAs are used by multiple *Cuscuta* species. Target predictions show that *C. campestris* homologs of targeted *A. thaliana* mRNAs invariably have lower complementarity to HI-sRNAs (*Figure 4*). Repeating the analysis pipeline to examine possible self-targeting of *C. campestris* mRNAs by HI-sRNAs found only four confirmed targeting interactions, an indication that HI-sRNAs may largely function in trans in the host (*Figure 4—figure supplement 1*).

Some mRNAs were confirmed to be targeted by HI-sRNAs from multiple species, with the most frequent interaction being with *SEOR1* (*Figure 2F*, *Figure 2—figure supplement 1A*). *SEOR1* encodes a phloem protein that acts to reduce sap loss after wounding (*Knoblauch et al., 2014*). *C. campestris* growth is enhanced when *A. thaliana seor1* mutants are used as hosts (*Shahid et al., 2018*). However, the majority of mRNAs confirmed as targets are unique to a single *Cuscuta* species or isolate (*Figure 2F*, *Figure 2—figure supplement 1A*). A possible explanation could be that *Cuscuta trans*-species sRNAs function to regulate similar host processes, while not necessarily the same target mRNAs. Additionally, our current analysis is likely to have missed many targets, both due to lack of sensitivity of our methods (secondary siRNA accumulation and/or degradome analysis both can miss true targets), and because *A. thaliana* is unlikely to be a major host of *Cuscuta* in nature.

Numerous target mRNAs are known to be involved in the same processes, both on a gene ontology level (*Figure 2—figure supplement 2*) and when manually examining known pathways. Genes involved in auxin signaling repeatedly appear, including the previously identified targets *TIR1*, *AFB2*, and *AFB3* (*Shahid et al., 2018*) and new targets *PXY* (*Etchells et al., 2012*) and *ARK2* (*Sankaranarayanan and Samuel, 2015*) with a proposed role in auxin response. Auxin signaling is involved in many processes in the plant, and is potentially connected to defense against *Cuscuta* through its role in glucosinolate production (*Smith et al., 2016*; *Salehin et al., 2019*). Phloem protein mRNAs are targeted, adding *OPS* (*Truernit et al., 2012*) to the previously identified *SEOR1*. A receptor-like kinase (*CuRe1*) from tomato is a resistance gene that prevents *Cuscuta reflexa* infestation (*Hegenauer et al., 2016*). Receptor-like kinases and kinases in general are well represented in the set of HI-sRNA targets, including several that are involved in defense responses. This includes the well-known defense regulator *MPK3* (*Asai et al., 2002*) and previously discovered *BIK1*

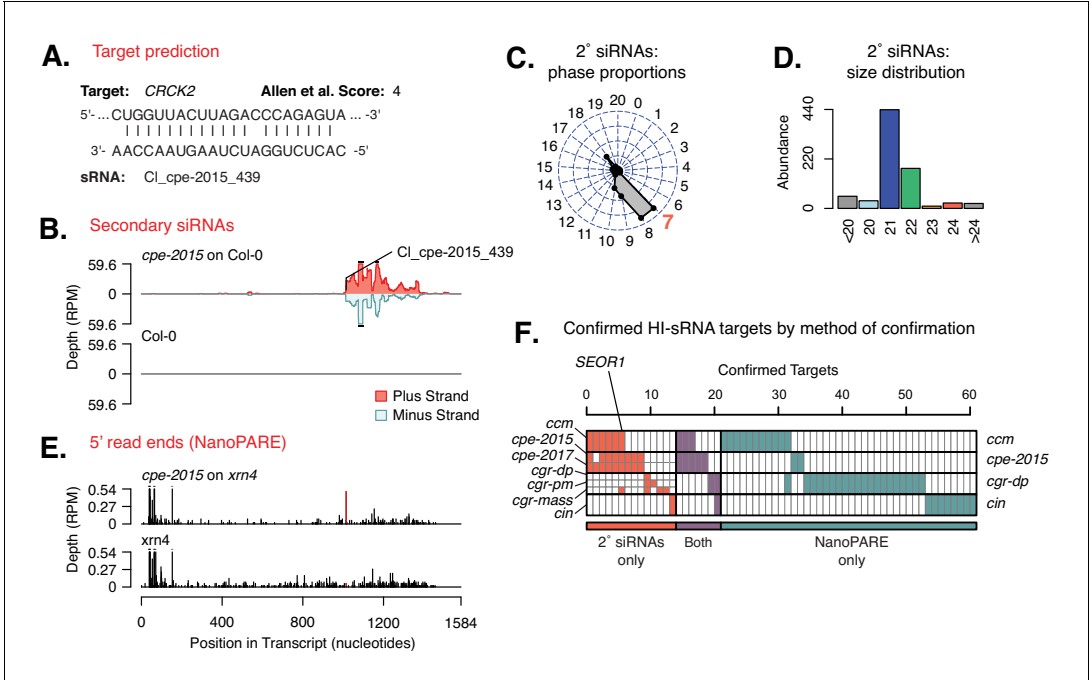

**Figure 2.** Host targets of *Cuscuta* HI-sRNAs. (**A**) Modeled sRNA-target interaction for *A. thaliana CRCK2*. (**B**) Secondary siRNA accumulation from *CRCK2*. (**C**) Phasing analysis of secondary siRNAs from *CRCK2*. Expected phase for cut-site shown in red. (**D**) Size distribution of *CRCK2* secondary siRNAs. (**E**) Frequency of 5′ ends from the *CRCK2* mRNA, with the predicted HI-sRNA cut site shown in red. (**F**) Host mRNAs with confirmed targeting by a *Cuscuta* HI-sRNA. Full details in *Figure 2—figure supplement 1* and *Supplementary file 6*.

The online version of this article includes the following figure supplement(s) for figure 2:

**Figure supplement 1.** Summary of *Cuscuta* HI-sRNA and host gene target relationships.

**Figure supplement 2.** Most common GO terms for confirmed target genes.

(*Veronese et al., 2006*). Another targeted pathway is brassinosteroid (BR) signaling, with targets *BRI1* (*Planas-Riverola et al., 2019*), *MAPKKK5* (*Yan et al., 2018*), and *PICKLE* (*Zhang et al., 2014*). BR has a clear role in defense, with connections to both *BIK1* and *MPK3* (*Zheng et al., 2018*). An overall theme of targeting host immunity and vascular system function emerges from this set of confirmed targets.

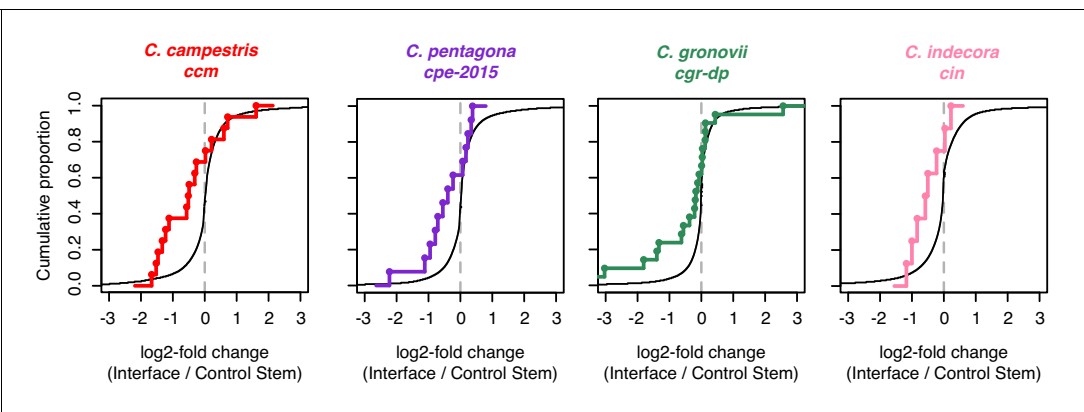

**Figure 3.** Analysis of mRNA accumulation in host-parasite interfaces. Cumulative density plots of interface/control stem ratios for host mRNAs expressed in *Cuscuta*-host interfaces, assessed by RNA-seq. All mRNAs shown with black line. Colored lines and dots indicate mRNAs which are confirmed targets of HI-sRNAs in the indicated *Cuscuta* isolates.

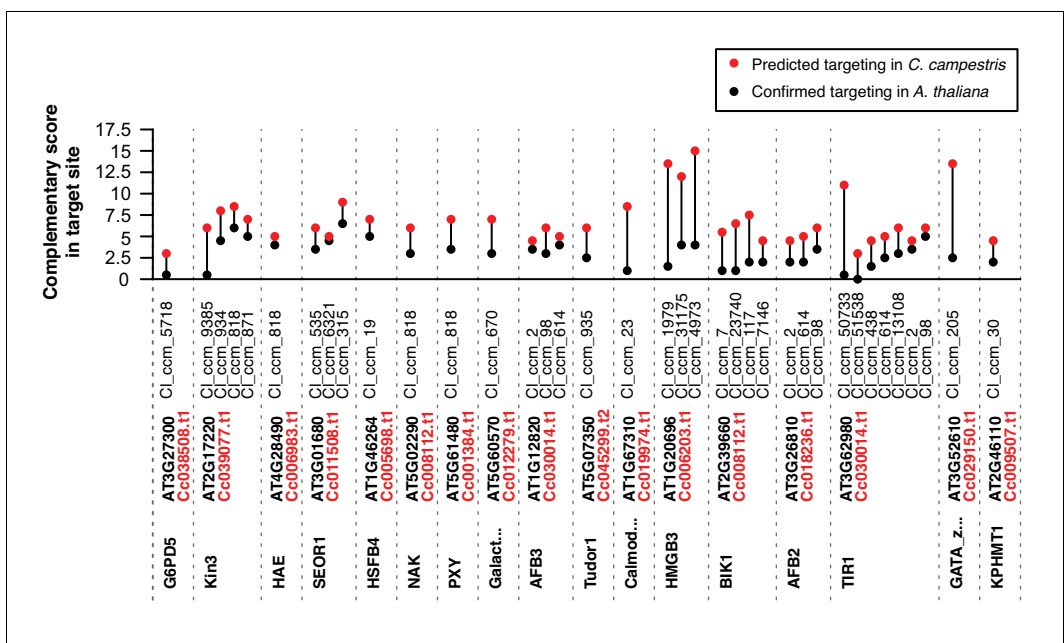

**Figure 4.** Predicted *trans*-species and self-targeting in *C. campestris* homologs of target *A. thaliana* mRNAs. Target prediction scores for confirmed *A. thaliana* mRNA targets (black) and best-blast-hit homologs in *C. campestris* (red). All sRNAs with predicted targeting are shown.

The online version of this article includes the following figure supplement(s) for figure 4:

**Figure supplement 1.** Experimental flowchart for confirming self-targeting of *C.campestris* mRNAs by HI-sRNAs.

Plant miRNAs that initially seem unrelated based on divergent sequences can sometimes be grouped into superfamilies (*Xia et al., 2013*). To discover potential superfamilies among *Cuscuta* HI-sRNAs, we clustered them with a cutoff of five substitutions and barring indels (*Figure 5—figure supplement 1*). This clustering strategy gives low rates of grouping by random chance (*Figure 5—figure supplement 2*). Many superfamilies of *Cuscuta* HI-sRNAs were found, with a substantial portion of them shared between species and isolates (*Figure 5A*). 19 superfamilies were shared between all isolates except *C. indecora*, and another 14 superfamilies were present in at least one isolate each of *C. campestris*, *C. pentagona*, and *C. gronovii*. Leveraging the prior *C. campestris* miRNA annotations, we can extrapolate that 158 out of 332 superfamilies which contain *C. campestris* HI-sRNAs are likely miRNAs. Furthermore, we extrapolate that of the superfamilies present in *C. campestris* with proven target relationships, 22 out of 23 are likely to be miRNAs (*Figure 2—figure supplement 1B*).

HI-sRNAs within a superfamily vary at several positions both between and within species (*Figure 5B*). In many cases variation within superfamilies occurred in a visible three-nucleotide period (*e.g.* SupFam_24, SupFam_37; *Figure 5B*, *Supplementary file 7*). This pattern led us to investigate nucleotide variation in corresponding target sites among possible hosts. All four *Cuscuta* species in this study are generalists that parasitize eudicot hosts, so we aligned homologous target mRNAs from 36 eudicot species (*Supplementary file 8*). Analysis of translated target site conservation shows that HI-sRNAs target highly-conserved protein-coding positions (*Figure 5C*). Positional variations in HI-sRNA superfamilies precisely correspond to variable positions in homologous target sequences (*Figure 5B*). This variation is frequently apparent at synonymous sites, accounting for the three-nucleotide periodicity of superfamily variation. Modeling correlation of positional variation between HI-sRNA superfamilies and eudicot target sites found 18 significant (p-value<0.05, Pearson correlation) examples of this type of co-variation (*Supplementary file 7*, *Figure 2—figure supplement 1C*). Importantly, HI-sRNA superfamily variation occurs within single *Cuscuta* species (*Figure 5B*, *Supplementary file 7*), such that multiple HI-sRNA variants are commonly deployed by a given parasite during infestation. By targeting conserved sites, and making several HI-sRNA

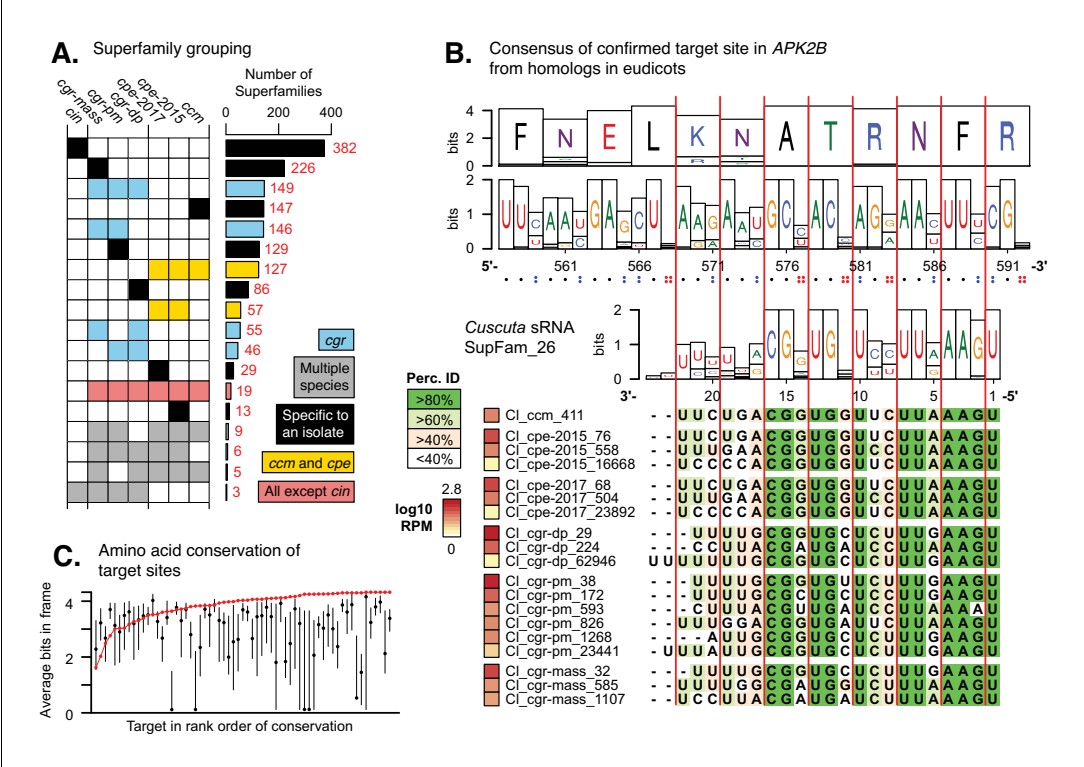

**Figure 5.** *Cuscuta* HI-sRNAs form superfamilies that co-vary with target sites across eudicots. (A) sRNA superfamily count and membership for each *Cuscuta* isolate. Colors indicate general groupings of superfamilies. (B) An example HI-sRNA superfamily aligned to target sites from homologs in 36 eudicot genomes. Nucleotide and amino acid Shannon entropy from the alignments are shown as bits. Vertical red lines indicate the frame. Dots indicate the number of possible synonymous nucleotides at each codon. 17 additional examples in *supplementary file 7*. (C) Average conservation of target sites from homologs. Confirmed target site shown (red point), with all other possible sites shown by 25–75% quartiles (black line) and median (black point).

The online version of this article includes the following figure supplement(s) for figure 5:

**Figure supplement 1.** Clustering method for forming HI-sRNA superfamilies.

**Figure supplement 2.** Testing distance cutoff parameters for superfamily formation.

variants that collectively cover many/most possible synonymous target variants, *Cuscuta* may ensure successful targeting across a wide range of hosts.

Using sRNA-seq libraries made from *C. campestris* attachments on *Nicotiana benthamiana* (*Shahid et al., 2018*), we found evidence of targeting in transcripts homologous to known *A. thaliana* targets (*Supplementary file 9*). Additionally, *N. benthamiana* target mRNAs were generally down-regulated in interface tissues (*Figure 6A*, *Supplementary file 10*. Comparing targeting of *TIR1* in *A. thaliana* and *N. benthamiana* homologs by SupFam_27 sRNAs illustrates differential complementarity of superfamily members to different mRNAs . The *N. benthamiana TIR1* target sites encode identical amino acids, but vary at synonymous positions. Some SupFam_27 variants are more complementary than others from each of the different homologs (*Figure 6B*). This provides a direct example where variation in a *Cuscuta* sRNA superfamily accommodates synonymous-site variation in confirmed target mRNAs from different plant species.

HI-sRNA superfamily diversity could also enable repression of multiple mRNAs with homologous target-sites within a single host. We examined target predictions within *A. thaliana* and found ten examples of gene family-specific motifs potentially targeted by *Cuscuta* HI-sRNA superfamilies (*Supplementary file 11*). These include a HI-sRNA superfamily predicted to target the mRNA region encoding the eponymous WRKY motifs within the well-known family of defense-related transcription factors (*Pandey and Somssich, 2009*). Several of the targets have been experimentally confirmed by secondary siRNA accumulation or degradome analysis but most remain predictions, including the

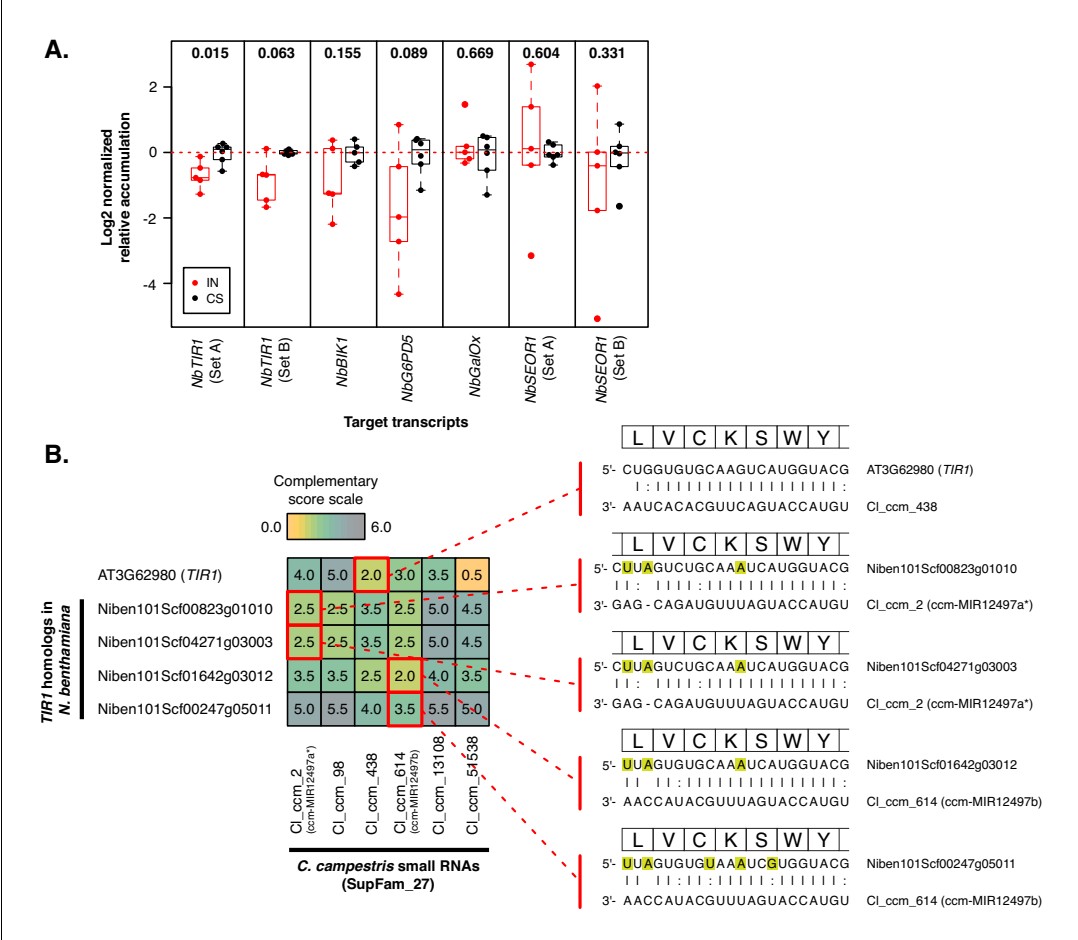

**Figure 6.** Superfamilies compensate for variation in *N.benthamiana* target homologs. (**A**) Accumulation of *N. benthamiana* target mRNAs. Interface (IN, red) and control stem (CS, black) are shown relative to average CS expression. Points represent biological replicates (N = 5 to 6). *P* values comparing IN to CS are displayed above the x axis; Wilcoxon rank-sum tests, unpaired, one-tailed. Accumulation was normalized to *NbTIP41-L* (Niben101Scf03385g06003) and *NbPP2A* (Niben101Scf09716g01002). (**B**) sRNA-target alignments of SupFam_27 sRNAs with *TIR1* family members from *N. benthamiana* and *A. thaliana*. Complementarity scores (*Allen et al., 2005*) are shown in the heatplot. The strongest predicted interactions are shown on the right; highlighted nucleotides are synonymous variants relative to *AtTIR1*.

*WRKY* family. However, false negatives are common with these methods of confirmation. The striking patterns of sequence covariation between the HI-sRNA superfamilies and their possible target mRNA families make a strong case for the reality of these interactions.

## Discussion

We conclude that multiple *Cuscuta* species use *trans*-species HI-sRNAs to target a substantial number of host mRNAs. Many if not most of these HI-siRNAs are likely to be miRNAs. Host genes involved in pathogen defense, hormone signaling, and vascular system function are common targets of *Cuscuta trans*-species HI-sRNAs. *Cuscuta trans*-species HI-sRNAs nearly always interact with highly conserved target sites within the coding sequences of host mRNAs. HI-sRNAs can often be grouped into superfamilies that have nucleotide diversity that corresponds with target site variation primarily at synonymous sites. It seems likely that host target sites are under purifying selection because they code for critical amino acids that have little variation even among distantly related eudicots. By targeting these already constrained protein-coding sites, and deploying an array of sRNA variants that cover most possible permutations of synonymous site variation, *Cuscuta* HI-sRNAs are likely to be robust against the evolution of host resistance by target site sequence changes. This is also a

suitable strategy for a generalist parasite that interacts with diverse hosts. The strategy used by *Cuscuta* provides a novel paradigm for the molecular evolution of *trans*-species sRNA targeting during parasitism.

# Materials and methods

**Key resources table**

| Reagent type (species) or resource | Designation | Source or reference | Identifiers | Additional information |
|---|---|---|---|---|
| Genetic reagent (*A. thaliana*) | *xrn4* | *Rymarquis et al., 2011* | *xrn4*-5; CS68822; SAIL_681_E01 | T-DNA insertion mutation in Col-0 background |
| Commercial assay or kit | Nextera DNAflex kit | Illumina | Product: 20018704 | |
| Commercial assay or kit | NEB primers set 1 | New England Biolabs | Product: E7335S | |
| Commercial assay or kit | NEB primers set 2 | New England Biolabs | Product: E7500S | |
| Commercial assay or kit | NEB primers set 3 | New England Biolabs | Product: E7710S | |
| Commercial assay or kit | NEB primers set 4 | New England Biolabs | Product: E7730S | |
| Software, algorithm | ShortStack | (*Johnson et al., 2016*) | v3.8.5 | https://github.com/MikeAxtell/ShortStack |
| Software, algorithm | DESeq2 | (*Love et al., 2014*) | v1.24.0 | https://bioconductor.org/packages/release/bioc/html/DESeq2.html |
| Biological sample (*C. campestris*) | ccm | *Shahid et al., 2018*; Jim Westwood, Virginia Tech | 'doddi' | |
| Biological sample (*C. pentagona*) | cpe-2017 | Ebay, seller: eden_wilds | 2017 collection | |
| Biological sample (*C. pentagona*) | cpe-2015 | Ebay, seller: eden_wilds | 2015 collection | |
| Biological sample (*C. gronovii*) | cgr-dp | Claude dePamphilis, Penn State | | Provenance unknown |
| Biological sample (*C. gronovii*) | cgr-mass | Jim Westwood, Virginia Tech | massachusetts isolate | Origin: A Massachusetts cranberry bog |
| Biological sample (*C. gronovii*) | cgr-pm | Wild collection | purdue mountain isolate | Origin: Roadside near State College, PA (Coordinates: 40.866 N, 77.888 W) |
| Biological sample (*C. indecora*) | cin | www.ars-grin.gov | PI 675068 | Origin: Texas |

## Seed sources

*Cuscuta campestris* (isolate 'doddi') was originally acquired from a tomato field in California, followed by several generations of selfing in the Westwood laboratory and provided to us as a gift. *C. gronovii* (isolate 'DP') was a gift C. dePamphilis, from an unknown source. *C. gronovii* (isolate 'mass') was collected in Massachusetts and provided as a gift by J. Westwood. *C. gronovii* (isolate 'PM') was collected from a road-side near State College, PA by M. Axtell in October 2017 (Coordinates: 40.866 N, 77.888 W). *C. pentagona* (isolates 'eden-2015' and 'eden-2017') were purchased from ebay seller eden_wilds in 2018, both collected from locations in upstate New York. *C. indecora* (isolate *cin*/GRIN) was acquired from the U.S. national plant germplasm system (www.ars-grin.gov) under the accession: PI 675068.

## Genotyping *Cuscuta*

*Cuscuta* seed were scarified by nicking with a razor blade under dissecting microscope and germinated on wet paper towel under growth lighting at ~28°C, harvesting seedlings after 3–5 days of growth for DNA extraction. DNA was extracted using Edwards method (*Kasajima et al., 2004*). 2 uL of template was used in a 20 uL PCR reaction using *Taq* polymerase using 0.5 uM final concentration of each primer (Primers 'c' and 'f', *Supplementary file 12*) (*Taberlet et al., 1991*). PCR was performed for 30 cycles and enzymatically cleaned-up using 0.5 uL of Exo1 (NEB) and 1 uL of Antarctic Phosphatase (NEB) with 5 uL PCR product, followed by incubation (15 min at 37°C) and inactivation (15 min at 80°C). Sanger sequencing was performed by the Penn State genomics core using primer 'c' (*Taberlet et al., 1991*). Sequences were trimmed of low quality bases and aligned using MUSCLE (*Edgar, 2004*) to published TrnL-F sequences (*Stefanovic et al., 2007*; *Costea et al., 2015*). Nucleotide phylogeny was constructed using MEGA7 (*Kumar et al., 2016*) with a maximum likelihood method and 500 bootstrap replicates (*Supplementary file 13*).

## Growth conditions

Host *A. thaliana* (Col-0 and *xrn4*) was sown on wet potting medium, followed by with 3 days of stratification at 4°C. Plants were placed into long day (16 day/8 night) growth conditions at ~23°C under cool-white-fluorescent lamps. Hosts were allowed to grow to maturity (4–5 weeks old), ready for attachment when first inflorescences were longer than 5 cm.

Cuscuta seeds were scarified and germinated as above. Seedlings were ready for attachment once completely emerged from their seed husk and roughly ~2 cm in length, 3–5 days depending on the species. Seedlings were placed in soil next to the primary bolts of host plants. House-built far-red supplementary LED lighting was used to induce attachment under fluorescent lights, allowing 4–5 days for attachment of the parasite. Once attached, parasitized hosts were removed from far-red lighting to prevent secondary attachment. For *C. indecora*, experimental attachments came from tendrils from a previously established *C. indecora* colony. 5 cm tendril tips were cut off of the colony and affixed to primary bolts of host plants with scotch tape. Plants were allowed to grow for 10 days after attachment followed by tissue harvest.

## Tissue collection and RNA extraction

All tissues were collected by the following methods and immediately submerged in liquid nitrogen to preserve RNA stability. Guide to tissues gathered is found in *Supplementary file 2*. Interface (IN) tissue was collected by taking both the host and parasite portions of the interface, trimming away any stems above and below the connection. Parasite stem (PS) was harvested ~4 cm above the interface, approximately 4 cm long each. In NanoPARE experiment using *xrn4* as a host, we collected only host interface (HIN); similar method to interface collection, except removing any parasite tissue which can be pulled away. Control stem (CS) tissues were harvested from non-parasitized *A. thaliana*, collecting stems from the same region where *Cuscuta* would have been attached. 1–3 tissues were pooled for each biological replicate. RNA was extracted by grinding tissue in a liquid nitrogen cooled mortar, with Tri-reagent (Sigma) added while still cold. Tri-reagent extraction was performed as per the manufacturer's suggestions with a second sodium-acetate–ethanol precipitation and wash step.

## Sequencing library preparation

All small-RNA-seq libraries were prepared using a protocol based on the NEBnext small-RNA library kit (NEB), described as follows. (**Step 1**) 3′ SR Adaptor (NJ410) was pre-adenylated using 5′ adenylation kit (NEB) as per manufacturer's instructions. (**Step 2**) 500 ng of total RNA, 1 μL adenylated adapter (5 μM) and water to 5.25 μL were denatured for 2 min at 70°C and immediately moved to ice. Entire reaction was combined with premixed 100 u RNA Ligase 2, truncated KQ (NEB), 1 μL 10x T4 RNA reaction buffer, 10 u RNAse inhibitor and 3 μL 50% PEG8000, to a total volume of 10 μL and incubated 1 hr at 25°C. (**Step 3**) Primer hybridization was performed adding 0.5 μL SR RT primer (NJ391, 10 μM) and 2.25 μL water to the prior reaction and incubated as follows: 5 min at 75°C, 15 min at 37°C, 15 min at 25°C, and holding at 4°C. (**Step 4**) 5′ SR RNA adaptor (NJ411) was diluted to 10 μM and denatured for 2 min at 70°C, moved to ice and used immediately for ligation. Ligation was performed combining the prior reaction with 0.5 μL denatured adapter (NJ411), 5 u RNA Ligase 1 (NEB), 0.25 μL 10x RNA ligase buffer, 10 u RNAse inhibitor, 0.5 μL ATP (10 mM), and water to 15 uL and incubated for 1 hr at 25°C.

(**Step 5**) Reverse transcription was performed immediately following ligation, combining the prior reaction with 100 u Protoscript II reverse transcriptase (NEB), 4.5 µL 5x first strand synthesis buffer, 1 µL dNTPs (10 uM), 1.5 µL DTT (0.1 M), and 10 u RNAse inhibitor equaling 23 µL in total volume and incubated for 1 hr at 50°C followed by heat-killing for 15 min at 70°C. (**Step 6**) Library amplification was performed, combining 5 µL of cDNA with 25 µL LongAmp Taq 2x master mix (NEB), 1.25 µL SR primer (NJ412, 10 µM), 1.25 µL barcode primer ('NEB' primers, 10 µM), and water to 50 µL total volume. Reaction was performed as follows: 30 s initial denature at 94°C, 15 cycles of 15 s at 94°C, 30 s at 62°C, and 15 s at 70°C, followed by final extension of 5 min at 70°C. Reactions were purified and size selected for sRNAs 15–40 nt in length by PAGE. Extracted bands were quantified by qPCR and quality-controlled by high-sensitivity DNA chip (Agilent). Sequencing was performed on a NextSeq550 (Illumina) with the high-output kit (75 nt, single-end, single barcode) by the Penn State genomics core. Sequencing libraries were de-multiplexed and adaptor trimmed using cutadapt (*Martin, 2011*) (cutadapt -a AGATCGGAAGA -m 15 j 8 -o *output.fq input.fq*).

NanoPARE and mRNA-seq libraries were prepared using the protocol described in *Schon et al. (2018)*, with the following details: NanoPARE and mRNA-seq were performed on interfaces (IN) of four isolates (*ccm*, *cpe-2015*, *cgr-dp*, and *cin*) and control stems (CS), grown on Col-0 *A. thaliana*. NanoPare was also performed on host interfaces (HIN) of the same isolates and control stem grown on *xrn4* mutant Col-0 *A. thaliana*. The Nextera DNA flex kit (Illumina) was used for tagmentation of 110 ng pre-amplified PCR product. Libraries were amplified using different barcoded i7 and i5 primer sets, described in *Supplementary file 12*), allowing for either amplification of 5' ends (Nano-PARE) or all tagged entities (mRNA-seq). The sequencing of NanoPARE data made use of custom read one sequencing and i5 index sequencing primers (NJ395 and NJ416, reverse complements of each other), which sequence out from the template switching oligo adapter. Sequencing was performed on a NextSeq550 (Illumina) with the high-output kit (75 nt, single-end, double barcoded) by the Penn State genomics core. Sequencing libraries were de-multiplexed and NanoPARE libraries were trimmed using an in-house script to remove any residual untemplated 5' nucleotides caused by reverse transcription of the template-switching oligo.

## Genome-free sRNA discovery

Genome-free sRNA discovery was performed using a set of in-house scripts, corresponding to the following pipeline (*Figure 1—figure supplement 2*). Reads were filtered by size, retaining lengths of 20 to 24 nt, and condensed to unique sequences with a count of abundances for each tissue. For each *Cuscuta* species, unique reads were further condensed by sequence similarity to their most abundant variants. This process found similar variants for sRNAs in rank order of abundance, clustering sRNAs with a Levenshtein edit distance of 2 or less. Reads which do not cluster to a variant with abundance of 0.5 reads per million (RPM) or higher are discarded. Most abundant sRNA variants of each cluster are reported, with the abundance as the combined abundance of all clustered reads. Host sRNAs were then filtered, removing an sRNA if it met one of the following criteria: (**1**) it is closely similar to an annotated miRNA; (**2**) it aligns perfectly to the *A. thaliana* genome or transcriptome (*Cheng et al., 2017*); (**3**) it is present in non-parasitized *A. thaliana* control libraries at an RPM greater than 1/100 its expression in parasite libraries. Differential expression analysis was then performed with DEseq2 (*Love et al., 2014*) to identify sRNAs up-regulated in the interface tissue relative to the parasite stem, using the command the 'results' command with a false-discovery rate of 0.1 (Benjamini–Hochberg correction). This pipeline resulted in our list of HI-sRNAs.

Superfamilies were constructed using an in-house script that corresponds to the following pipeline. All by all comparisons of HI-sRNA sequences were performed, measuring modified hamming distance (*Figure 5—figure supplement 1*), and sequences with a distance of 5 or less were clustered together, ordered by overall size of the superfamily. To test this distance cutoff, HI-sRNA sequences were shuffled using Ushuffle (*Jiang et al., 2008*), set to retain di-nucleotide structure (10 random replicates) (*Figure 5—figure supplement 2*).

## Target confirmation

Target prediction of HI-sRNAs was performed using the script GSTAr.pl (https://github.com/MikeAxtell/GSTAr; *Axtell, 2014*; copy archived at https://github.com/elifesciences-publications/

GSTAr) under default settings, using HI-sRNA from a given isolate as the query and *A. thaliana* ARA-PORT11 transcriptome (*Cheng et al., 2017*) as the subject.

To find secondary siRNAs produced from targeting of *A. thaliana* genes, sRNA-annotation was performed on the *A. thaliana* genome (*Arabidopsis Genome Initiative, 2000*), using ShortStack (*Johnson et al., 2016*) (https://github.com/MikeAxtell/ShortStack) with gene locations from the ARAPORT11 annotation (*Cheng et al., 2017*) as the basis for sRNA loci. Differential expression analysis was performed with DEseq2 (*Love et al., 2014*) to identify loci up-regulated in the interface (IN) relative to control stem (CS), using the 'results' command with a false-discovery rate of 0.1 (Benjamini–Hochberg correction). Up-regulated loci were then filtered to retain loci which met the following criteria: (**1**) have a strong predicted HI-sRNA target site (complementarity score (*Allen et al., 2005*) six or less); (**2**) are unstranded; (**3**) have a predominant sRNA length of 21/22 nt; (**4**) have a minimum depth of 20 reads. Plots of loci with predicted targets, radar plots of sRNA phasing, and length distribution plots to find examples where HI-sRNAs are clearly causative in the locus.

To confirm targeting of *A. thaliana* genes using degradome data, host stem of interface (HIN) NanoPARE libraries were aligned to the ARAPORT11 (*Cheng et al., 2017*) transcriptome using bowtie (*Langmead et al., 2009*) (bowtie -p 8 f -v 3 S -a). Using an in-house script, frequency at 5' positions of alignments were intersected with predicted HI-sRNA target sites, retaining interactions which met the following criteria: (**1**) have a strong prediction score (complementarity score six or less; *Allen et al., 2005*) six or less; (**2**) the target site is greater than 100 nt from the start of the transcript (to avoid miscalls with the transcriptional start site); (**3**) the target is not in an organellar genome; (**4**) the target peak is greater than the median peak depth in the gene; (**5**) the target peak is found in all three replicates; (**6**) the target peak is at least 10 fold higher than detected in control stems. Candidates were then examined by eye, filtering out hits with low expression compared to the rest of the gene, hits which appear to be in the transcriptional start site, or hits with low prominence compared to surrounding peaks. Gene-ontology analysis was performed on confirmed targets using blast2go (*Götz et al., 2008*).

Confirmation of targeting in *C. campestris* genes (*Figure 4—figure supplement 1*) was performed using the similar methods as above, with the following changes. Secondary siRNAs were annotated with ShortStack (*Johnson et al., 2016*) to the *C. campestris* genome (*Vogel et al., 2018*), using gene annotations as the basis for loci. Different NanoPARE libraries were used, coming from mixed host-parasite interface (IN), and were aligned to the *C. campestris* transcriptome (*Vogel et al., 2018*). No direct control was present to compare peak expression, so the few confirmed examples could not be subjected to this filter.

## mRNA-seq analysis

mRNA-seq libraries were aligned to the *A. thaliana* genome (Arabidopsis Genome Initiative, 200) using HISAT2 (*Kim et al., 2015*) (hisat2 -p 2 –max-intronlen 5000 -x genome.fa -U library.fq.gz). Gene expression was quantified by minBamCov (*Barnett et al., 2011*) (multiBamCov -bams alignment.bam -bed annotation.gff) using the ARAPORT11 annotation (*Cheng et al., 2017*). Deseq2 (*Love et al., 2014*) was used to accurately estimate log fold change of mRNAs for each condition, using the 'lfcShrink' command (type = apeglm).

## Identification of *C. campestris* miRNAs

To identify sRNAs that were derived from miRNA hairpins, de novo annotation of sRNA loci in the *C. campestris* genome (*Vogel et al., 2018*) was performed using ShortStack (*Johnson et al., 2016*). Next, loci containing a HI-sRNA from *C. campestris* were extracted and screened by eye to find miRNAs with the criteria that they have a clear concise hairpin with two matching regions of expression which have a clear two nt offset (factors consistent with miRNA processing). Superfamilies were annotated to identify which contained confirmed miRNAs.

## Discovery of target homologs in eudicots

cDNA and CDS libraries of 36 eudicot species (*Supplementary file 8*) available in Phytozome v12.1.6 (*Goodstein et al., 2012*) were downloaded for local analysis. Nucleotide queries of *A. thaliana* target transcripts were searched against translated CDS libraries from eudicots using blastx (*Camacho et al., 2009*) (blastx -query target.fa -db eudicot.db -outfmt 6 -num_threads 6 -evalue

0.001 -task blastx-fast), extracting the best hit for each species based on bit score. Conservation of target site and coding sequence of homologs was calculated by aligning their translated coding sequences using MUSCLE (*Edgar, 2004*) and measuring the average conservation (Shannon entropy) of every eight amino acid window, flagging the window which corresponds to the target site. RNA superfamilies and transcripts of best-hit homologs of target were each aligned using MUSCLE (*Edgar, 2004*) and oriented to each other using in-house scripts. Positional nucleotide conservation for target site in homologs and superfamily sRNAs were calculated and used to construct a linear regression model in R (lm function and resulting p-values). For each comparison, n equals the number of correlating nucleotide positions in the interaction (n = 20–24).

## Discovery of conserved motifs in *A. thaliana* targets

Conserved motifs targeted by sRNA superfamilies in *A. thaliana* were found by first extracting all targets of a superfamily with very strong predicted targeting (complementarity score [*Allen et al., 2005*] three or less). Using an in-house script, sequences of target sites were translated for the correct frame and clustered using a greedy algorithm with a maximum edit distance in a cluster of three or less. Conservation of target sites and surrounding nucleotide sequences were then calculated and oriented adjacent to multiple sequence alignments of the targeting superfamily, highlighting confirmed interactions.

## Target analysis in *N. benthamiana*

sRNA sequencing libraries of *C. campestris* parasitizing *N. benthamiana* were retrieved from SRA bioproject: PRJNA408115 (*Supplementary file 2*) (*Shahid et al., 2018*). Parasite stem and interface libraries were used as input in the genome-free sRNA discovery pipeline explained above. Secondary siRNA producing loci were identified using the pipeline explained above with the *N. benthamiana* Genome v1.0.1 (*Bombarely et al., 2012*). Transcripts that were identified as containing HI-sRNA induced secondary siRNA loci were then compared to *C. campestris* HI-sRNA targets in *A. thaliana*, identifying possible homologs of these targets (*Supplementary file 10*).

For the quantification of target mRNA expression, *C. campestris* was attached to 2–3 week old *N. benthamiana* plants, and allowed to grow for 11–15 days. Total RNA was extracted from the interface of parasitized (IN) and the stem of unparasitized plants (CS), using Tri-reagent (Sigma) and the double-precipitation method explained above. cDNA was synthesized from 1 µg of total RNA using the ProtoScript II (NEB) reverse transcriptase with random primers, as per manufacturer instructions. qRT-PCR primers (*Supplementary file 12*) were designed for these transcripts, sometimes targeting several homologous mRNAs in *N. benthamiana* equally well. The amplicons were designed to bridge the best predicted cut sites among causative sRNAs. Best-blast-hit homologs of housekeeping genes PP2A (AT1G13320) and TIP41-L (AT4G34270) in *N. benthamiana* were used for normalization of PCR accumulation between cDNAs, averaging expression relative to control stem. qPCR resulted in expression from 5 to 6 biological replicates in IN and CS samples.

## Code availability

ShortStack (*Johnson et al., 2016*) and StrucVis are both freely available at https://github.com/MikeAxtell/strucVis (*Axtell, 2018*; copy archived at https://github.com/elifesciences-publications/strucVis). GSTAr.pl is freely available at https://sites.psu.edu/axtell/software/misc-tools/. Muscle (*Edgar, 2004*) is freely available at https://www.drive5.com/muscle/. MEGA7 (*Kumar et al., 2016*) is freely available at https://www.megasoftware.net/. Blast-suite (*Camacho et al., 2009*) is freely available at https://blast.ncbi.nlm.nih.gov. Cutadapt (*Martin, 2011*) is freely available at https://cutadapt.readthedocs.io/en/stable/. Bamtools (*Barnett et al., 2011*) is freely available at https://bedtools.readthedocs.io/en/latest/index.html#. The R package DEseq2 (*Love et al., 2014*) is freely available at https://bioconductor.org/packages/release/bioc/html/DESeq2.html. HISAT (*Kim et al., 2015*) and bowtie (*Langmead et al., 2009*) are both freely available at https://ccb.jhu.edu/software. Ushuffle (*Jiang et al., 2008*) is freely available at https://github.com/guma44/ushuffle. Blast2go (*Götz et al., 2008*) is available with a limited free version at https://www.blast2go.com/.

## Data availability

sRNA-seq data from this work are available at the NCBI SRA under BioProject PRJNA543296.

## Acknowledgements

We thank T Phifer for help generating preliminary data; LS Berghard for greenhouse support; C Praul for next-generation sequencing support; J Westwood (Virginia Tech) for providing multiple *Cuscuta* seed stocks and thoughtful insight into the work; C Depew for informing us about *C. gronovii* locations in the wild; and M Schon and M Nodine for early access to the NanoPARE method. This work was supported by an award from the United States Department of Agriculture - National Institute of Food and Agriculture [grant number 2018-67013-285] and a Graduate Research Initiative grant (GRI) from the Huck Institutes of the Life Sciences at Penn State.

## Additional information

### Funding

| Funder | Grant reference number | Author |
|---|---|---|
| National Institute of Food and Agriculture | 2018-67013-28514 | Michael J Axtell<br>Claude W dePamphilis |

The funders had no role in study design, data collection and interpretation, or the decision to submit the work for publication.

### Author contributions

Nathan R Johnson, Conceptualization, Data curation, Formal analysis, Investigation, Methodology; Claude W dePamphilis, Conceptualization, Funding acquisition, Methodology; Michael J Axtell, Conceptualization, Supervision, Funding acquisition, Methodology, Project administration

### Author ORCIDs

Michael J Axtell (iD) https://orcid.org/0000-0001-8951-7361

### Decision letter and Author response

Decision letter https://doi.org/10.7554/eLife.49750.sa1
Author response https://doi.org/10.7554/eLife.49750.sa2

## Additional files

### Supplementary files

• Supplementary file 1. Unabridged phylogeny of *Cuscuta* Phylogeny based on TrnL-F sequencing using vouchered samples and primers (*Stefanovic et al., 2007*; *Costea et al., 2015*).Isolates used in this study are in bold and indicated with arrows. Samples identified as members of species examined in this study are highlighted with color; red - *C. campestris*, purple - *C. pentagona*, green - *C. gronovii*, pink - *C. indecora*. Format: PDF

• Supplementary file 2. List of all libraries and tissues prepared or used in this study.All libraries are available under the SRA BioProject: PRJNA543296. Format: xlsx

• Supplementary file 3. Testing alternative p-value cutoffs for HI-sRNA detection.Format: xlsx

• Supplementary file 4. Comprehensive list of haustorium-induced small RNAs (HI-sRNAs) discovered in this study.Format: xlsx

• Supplementary file 5. Predicted secondary structures of miRNA hairpins producing HI-sRNAs in *C. campestris*.Predicted RNA secondary structures and expression profiles of loci that produce HI-sRNAs and have an apparent miRNA hairpin. Format: PDF

• Supplementary file 6. Target confirmation data for every confirmed HI-sRNA-target interaction. Details of confirmed HI-sRNA targets including HI-sRNA-target complementarity, site, score, superfamily and the status of *C. campestris* superfamily members as a confirmed miRNA. Targeting confirmation for target mRNA is shown in upper right, with confirmed interactions in species highlighted in red. sRNA distribution at target locus is shown for experimental interface and control,

demonstrating secondary siRNA phasing and size distribution for up-regulated loci. Degradome sequencing is shown where confirmed hits were discovered in NanoPARE data. Format: PDF

• Supplementary file 7. Target interactions with significant correlation of variation in superfamily and target site.Multiple sequence alignments of HI-sRNA superfamilies which have significant correlations between sRNA positional variation and target site variation. Alignment of eudicot homologs around target site also shown, with nucleotide and amino acid Shannon entropy shown as bits. Vertical red lines indicate the frame. Dots indicate the number of possible synonymous nucleotides at a position for the confirmed target's sequence. Nucleotide positions are in reference to the position in the multiple sequence alignment. Format: PDF

• Supplementary file 8. Eudicot genomic resources used in this study.All available in Phytozome version v12.1.6. Format: xlsx

• Supplementary file 9. Target confirmation data for every confirmed HI-sRNA-target interaction in N. benthamiana.Details of confirmed C. campestris HI-sRNA targets in N. benthamiana, including HI-sRNA-target complementarity, site, score, superfamily and the status of C. campestris superfamily members as a confirmed miRNA. sRNA distribution at target locus is shown for experimental interface and control, demonstrating secondary siRNA phasing and size distribution for up-regulated loci. Format: PDF

• Supplementary file 10. N. benthamiana targets of HI-sRNAs Based on N. benthamiana genome v1.0.1.Format: xlsx

• Supplementary file 11. Target interactions of A. thaliana homologs with conserved target motifs. Multiple sequence alignments sRNA of superfamilies and conserved target motifs found in Arabidopsis transcriptome, with nucleotide and amino acid Shannon entropy shown as bits. Vertical red lines indicate the frame. Dots indicate the number of possible synonymous nucleotides at a position for the confirmed target's sequence. Nucleotide positions are in reference to the position in the multiple sequence alignment. Color of gene names indicates if there is evidence for targeting in NanoPARE data (black - 0 replicates; orange - 1 or two replicates; red - three replicates, confirmed interaction). Format: PDF

• Supplementary file 12. List of primers used in this study.Format: xlsx

• Supplementary file 13. Alignment of TrnL-F sequences from Cuscuta.These were the basis for the phylogenetic tree presented in *Supplementary file 1*. Format: FASTA (plain text).

• Transparent reporting form

## Data availability

Sequencing data have been deposited in NCBI SRA under Bioproject number PRJNA543296.

The following dataset was generated:

| Author(s) | Year | Dataset title | Dataset URL | Database and Identifier |
|---|---|---|---|---|
| Johnson NR, Axtell MJ | 2019 | Small RNA-seq from multiple Cuscuta species parasitizing Arabidopsis | http://www.ncbi.nlm.nih.gov/bioproject/?term=PRJNA543296 | NCBI SRA, PRJNA543296 |

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
