## [Decision Letter]

**Acceptance summary:**

Small RNAs (sRNAs) produced in one species can move into another to regulate gene expression and interactions between species. It has been previously reported that the parasitic plant *Cuscuta* produces microRNAs (miRNAs) to silence host plant target mRNAs. In this study, the authors identified many more haustorially-induced sRNAs (HI-sRNAs) in several *Cuscuta* species and confirmed targets of some HI-sRNAs. A subset of the HI-sRNAs are shared between *Cuscuta* species and isolates and can be grouped into superfamilies. Interestingly, the authors found that HI-sRNAs within a superfamily have variation in a three-nucleotide period and their target mRNAs contain variations at the synonymous sites within the target sites. The authors propose that such co-variation may enable the HI-sRNAs to target homologous mRNAs in different host plants. These findings may have implications in the other parasite/pathogen-host interactions.

**Decision letter after peer review:**

Thank you for submitting your article "Compensatory sequence variation between trans-species small RNAs and their target sites" for consideration by *eLife*. Your article has been reviewed by three peer reviewers, one of whom is a member of our Board of Reviewing Editors, and the evaluation has been overseen by Christian Hardtke as the Senior Editor. The following individual involved in review of your submission has agreed to reveal their identity: Sascha Laubinger (Reviewer #2).

The reviewers have discussed the reviews with one another and the Reviewing Editor has drafted this decision to help you prepare a revised submission.

The reviewers agreed that such findings should in principle be published at the highest levels. While the reviewers furthermore agreed that the data presented are clear and thorough, they also agreed that some additional experiments would be essential to improve the manuscript.

Essential revisions:

Some experimental data are required to show that *Cuscuta* HS-sRNAs indeed use sequence variations to target homologous genes with corresponding sequence variations in different plant species or members within the same gene family in the same host plant. The following experiments are suggested.

1) Generate complementation lines of seor1, or apk2b, or other 1-2 of the 19 examples that contain synonymous mutations. The lines contain the mutated sequences in the synonymous sites but not cleaved by the Hi-sRNAs, and compare the infection results with the mutant and normal complementation lines in Arabidopsis. The reviewers realized that generating stable transgenic lines takes a long time and thus suggested the authors to provide results if they already have the lines.

2) Perform 5' RACE experiments in non-Arabidopsis plants infected by different *Cuscuta* species to confirm that the predicted targets are cleaved by HI-sRNAs.

3) Examine the cleavage of target sequences from different hosts by a few representative *Cuscuta* HI-sRNAs using reporter assays in tobacco leaves.

4) Perform qRT-PCR or RNA-seq analysis in Arabidopsis and other hosts infected by *Cuscuta* species to show that the target genes are indeed down-regulated by *Cuscuta* HI-sRNAs.

---

## [Author Response]

Essential revisions:Some experimental data are required to show that Cuscuta HS-sRNAs indeed use sequence variations to target homologous genes with corresponding sequence variations in different plant species or members within the same gene family in the same host plant. The following experiments are suggested.

Overall we've addressed this by adding new data and analyses that demonstrate targeting of homologous mRNAs, with synonymous sequence variants, from *Nicotiana benthamiana*. The newly added data are in new Figure 6, and Supplementary files 9 and 10.

1) Generate complementation lines of seor1, or apk2b, or other 1-2 of the 19 examples that contain synonymous mutations. The lines contain the mutated sequences in the synonymous sites but not cleaved by the Hi-sRNAs, and compare the infection results with the mutant and normal complementation lines in Arabidopsis. The reviewers realized that generating stable transgenic lines takes a long time and thus suggested the authors to provide results if they already have the lines.

We agree that these experiments would be informative. We have started producing the plant materials for these experiments, but they are not ready in time for this manuscript.

2) Perform 5' RACE experiments in non-Arabidopsis plants infected by different Cuscuta species to confirm that the predicted targets are cleaved by HI-sRNAs.

Confirmation of targeting in *Nicotiana benthamiana* was shown for *Cuscuta campestris* miRNAs by 5’ RACE and secondary siRNA identification in our prior publication (Shahid et al., 2018). To address this question in the context of this project, we re-analyzed the prior sRNA-seq data with our genome-free pipeline, confirming targeting for many of these transcripts through the presence of secondary siRNAs (now shown in Supplementary files 9 and 10). We show a more in-depth example of a sRNA superfamily targeting multiple TIR1 homologs in *N. benthamiana* (Figure 6), which illustrates a clear case where synonymous variants of a target (comparing *N. benthamiana* and *A. thaliana*) are compensated for by miRNA superfamily variation.

3) Examine the cleavage of target sequences from different hosts by a few representative Cuscuta HI-sRNAs using reporter assays in tobacco leaves.

Thank you for the suggestion. While potentially informative, we feel this experiment is outside of the scope of our present study. We do worry that artificial reporter assays might not reflect the true situation during parasite infestation, primarily because of the different modes of microRNA delivery.

4) Perform qRT-PCR or RNA-seq analysis in Arabidopsis and other hosts infected by Cuscuta species to show that the target genes are indeed down-regulated by Cuscuta HI-sRNAs.

Done. In the modified Figure 3 we show an analysis of confirmed targets based on mRNA-seq in Arabidopsis, confirming a general trend of target down-regulation during parasite infestation. We have also now performed qRT-PCR experiments from *C. campestris – N. benthamiana* samples for several confirmed targets. In general these *N. benthamiana* targets also trend toward down regulation, with the TIR1 example significantly down-regulated (newly added Figure 6).